# Knowledge, Attitudes and Practices of Eye Health among Public Sector Eye Health Workers in South Africa

**DOI:** 10.3390/ijerph182312513

**Published:** 2021-11-27

**Authors:** Zamadonda Xulu-Kasaba, Khathutshelo Mashige, Kovin Naidoo

**Affiliations:** 1Discipline of Optometry, School of Health Sciences, University of KwaZulu-Natal, Durban 4000, South Africa; mashigek@ukzn.ac.za (K.M.); NaidooK6@ukzn.ac.za (K.N.); 2Department of Optometry, University of New South, Wales Sydney, NSW 2052, Australia

**Keywords:** visual impairment, human resources for eye health, avoidable blindness, eye health, public health, eye health directorate

## Abstract

In South Africa, primary eye care is largely challenged in its organisational structure, availability of human and other resources, and clinical competency. These do meet the standard required by the National Department of Health. This study seeks to assess the levels of knowledge, attitudes, and practices on eye health amongst Human Resources for eye health (HReH) and their managers, as no study has assessed this previously. A cross-sectional study was conducted in 11 districts of a South African province. A total of 101 participants completed self-administered, close-ended, Likert-scaled questionnaires anonymously. Binary logistic regression analysis was conducted, and values of *p* < 0.05 were considered statistically significant. Most participants had adequate knowledge (81.6%), positive attitudes (69%), and satisfactory practices (73%) in eye health. HReH showed better knowledge than their managers (*p* < 0.01). Participants with a university degree, those aged 30–44 years, and those employed for <5 years showed a good attitude (*p* < 0.05) towards their work. Managers, who supervise and plan for eye health, were 99% less likely to practice adequately in eye health when compared with HReH (aOR = 0.012; *p* < 0.01). Practices in eye health were best amongst participants with an undergraduate degree, those aged 30–44 years (aOR = 2.603; *p* < 0.05), and participants with <5 years of employment (aOR = 26.600; *p* < 0.01). Knowledge, attitudes, and practices were found to be significantly moderately correlated with each other (*p* < 0.05). Eye health managers have poorer knowledge and practices of eye health than the HReH. A lack of direction is presented by the lack of adequately trained directorates for eye health. It is therefore recommended that policymakers review appointment requirements to ensure that adequately trained and qualified directorates be appointed to manage eye health in each district.

## 1. Introduction

Visual impairment is a serious public health problem globally. It is estimated that 253 million people worldwide are affected by visual impairment. In Sub-Saharan Africa (SSA), 22 million people are blind or visually impaired mainly from avoidable causes such as cataracts and uncorrected refractive errors [1]. Over 100 million adults in SSA are estimated to have near visual impairment [1]. Blindness from avoidable causes is said to have increased in all four regions of SSA in the past decade [2]. The age-standardised prevalence of blindness (>50 years) was found to be 5.1% in western and 4.3% in eastern SSA [1]. The disproportionate burden of visual impairment in low-and-middle-income countries (LMIC) compared to high-income countries was observed to be a direct cause of socioeconomic factors, poor health systems and concomitant human immunodeficiency virus (HIV), and tuberculosis epidemics [3,4,5,6,7]. The World Health Organization’s 2014–2019 global action plan (GAP) for universal eye health aimed to reduce avoidable vision loss, thereby curbing the quality-of-life limitations and economic demands associated with visual disabilities [8,9,10].

The World Health Organisation (WHO) has recommended that primary eye care (PEC) be included in primary healthcare (PHC) as a strategy to increase sustainability and access to ocular health services [11,12]. To effectively control visual impairment, the WHO highlighted the importance of accessible eye care services and called on member states to secure the inclusion of PEC within PHC, as previously recommended by the International centre for eye health [8,13]. Many challenges such as lack of agreement on the scope of PEC and lack of clear guidelines on the technical eye-related skills required by PHC workers were reported as challenges for the effective implementation of PEC in SSA. These affect the extent of training, supervision, and the type of equipment and consumables required [14].

In South Africa, PEC is mainly provided at the PHC level, but if need arises, patients are referred to higher-level institutions. The country does not have a dedicated directorate for eye health, nor does it have an integrated eye health promotional policy [15]. This results in inadequate eye care services, similar to other African countries [16,17]. Challenges in the South African eye care programme include insufficient human resources, unaffordable or unavailable medication, unsatisfactory programme evaluation and inadequate service coverage for Vitamin A supplementation, vision assessments, spectacle provision, cataract surgery, and screening for eye complications in patients with diabetes [18,19,20,21,22,23]. In addition, coordination between the different levels of the eye health system is lacking, with poor communication, a complex referral system and problems transporting patients to specialised services [19].

Studies from South Africa have reported on the prevalence of visual loss/visual impairment in different districts/provinces [24,25,26,27]. Another study performed an evaluation of primary eye care services in three districts of South Africa to assess whether an ophthalmic health system strengthening (HSS) package could improve these services [28]. The study concluded that primary eye care in South Africa faces multiple challenges with regard to the organisation of care, and clinical competency [28]. Training of all cadres of eye health was said to be crucial if the goals of VISION 2020 were to be attained, and universal access to ocular health achieved [29]. Very little is known about the knowledge, attitudes, and practices of eye health care workers and their supervisors towards eye health. Therefore, this study aimed to establish the level of knowledge, attitudes, and practices of eye health amongst HReH and their supervisors/managers. In this study, participants were tasked with responding to questions on the definitions of the different HReH, their roles in their work, resources needed in eye health, and challenges that exist in eye health daily. Policies guiding HReH work were also included in the questions.

Based on the responses, study findings will assist in clarifying the levels to which management and HReH each understand staffing roles and needs within the province, leading to possible interventions needed for optimal service provision. This study will also inform policymakers, healthcare administrators, and eye care professionals on areas that need attention in public health policies, further promoting efficient and equitable allocation of resources to alleviate the burden of vision loss in South Africa.

## 2. Methods

A cross-sectional study was conducted in the 56 eye clinics and 11 district offices in the province of KwaZulu-Natal, South Africa. The population for the study comprised two levels of managers, district office-based NCD coordinators and medical managers, who manage HReH within the various HCFs. Included HReH were Optometrists, Ophthalmologists, Ophthalmic Medical Officers (OMOs), and Ophthalmic Nurses, as well as an administrator, working in the various eye clinics.

Purposive sampling was used to identify 196 role-players within eye health in KwaZulu-Natal, a total population of 174 HReH and 22 Managers. Due to this sample size being relatively small, a sample size calculation was deemed to be irrelevant. In an attempt to obtain a saturated sample, the PI contacted all the eye clinics and made arrangements to personally visit each institution so as to ensure a saturated sample. Of the 196 eye health workers, 91 were either on leave, ill, occupied by other work, or unavailable for other reasons. The remaining 105 employees who were available all accepted the invitation to take part in the study. Ultimately, 101 eye health workers returned the self-administered, completed questionnaire after the allocated 20 min time frame, yielding a response rate of 96,2%.

The questionnaire comprised four sections. The first section of the questionnaire contained demographic information such as age, race, district, role in eye health, period of service, and highest level of qualification. The second, third, and fourth sections of the questionnaire comprised ten statements for each section to determine participants’ knowledge, attitudes, and practices on eye health. All the statements were 5-point Likert type with the categories ranging from “Strongly Disagree” to “Strongly Agree”.

The questionnaire was pretested among 10 HReH members who had resigned from the public sector eye clinics within two years prior to the commencement of this study. They responded to the questions and gave comments on the questionnaire. Amendments were made wherever needed, and the tool was modified and validated for this study. The Cronbach alpha scores were 0.72 for knowledge, 0.85 for attitude, and 0.84 for practices (Appendix A).

Ethical clearance for the study was granted by the University of KwaZulu-Natal (BE155/19) and the Department of Health Research Ethics Committee. Anonymity and confidentiality were maintained at all times. Participation in the study was voluntary.

Data were cleaned, coded, captured, and analysed using SPSS version 25. The Likert scale responses were condensed to elicit binary responses. Where the correct response was an agreement, “Strongly agree and Agree” were accepted as favourable responses while “Neutral, Disagree and Strongly disagree” were considered to be unfavourable. Similarly, where the correct response was a disagreement, “Strongly Disagree” and “Disagree” were accepted as favourable responses while “neutral”, “I don’t know”, “Agree” and “Strongly Agree” were rejected as unfavourable responses. Participants who correctly answered a minimum of 75% of the questions were considered to have adequate knowledge, a positive attitude, and satisfactory practicing skills.

## 3. Results

Most of the study participants were Africans (91%). About half (44.6%) were aged between 30 and 44 years, and HReH contributed 76.2% of the responses (Table 1). The highest qualification levels amongst the participants were a university degree (48.5%), a post-basic diploma in ophthalmic nursing (20.8%), a postgraduate degree (17.8%), a diploma (6.9%), and those with a grade 12 or a certificate for a short course were 6% were 6% of the study population.

### 3.1. Analysis of Knowledge

Table 2 shows a summary of the responses related to knowledge regarding eye health. Results show that the majority of the participants answered correctly to most of the statements. It was found that almost all the participants (95%) knew which eye health services were provided in their hospitals. An overwhelming majority of the participants agreed that an Ophthalmic Nurse provides the role of performing eye screening and assisting in theatre (83%), and an Optometrist is central in performing refraction and low vision services (87%), respectively. About two-thirds (65%) of the participants disagreed that an Optometrist is the HReH performs general primary eye health. Overall, 82% of the participants had good knowledge regarding eye health.

Table 3 shows the results from binary logistic regression analysis to determine the significant factors for having good knowledge. According to binary logistic regression analysis, HReH were 14 times more likely to have better knowledge (aOR = 14.21; *p* < 0.01) than their managers. Participants having a certificate qualification were 98% less likely to have good knowledge (aOR = 0.02; *p* < 0.05) compared to those with a higher level of education (a university degree and a postgraduate qualification). Respondents in the middle-aged (30–44) group were 12 times more likely to have better knowledge (aOR = 12.02; *p* < 0.01) than those in the oldest age group (>44 years).

Figure 1 reports the frequency distribution of the statements regarding attitudes towards eye health. It was found that most of the participants showed positive attitudes towards eye health. For example, 90% of the participants thought that Glaucoma, Diabetic Retinopathy, and Uncorrected Refractive Error should be treated as priority areas of care, and eye health is not about cataract surgery, which should be known to the directorate. Just over half of the participants agreed that the prevention of blindness should be prioritised, as most blinding conditions are preventable. Overall, 69% of the participants showed positive attitudes towards eye health.

Binary logistic regression analysis (Table 4) showed that participants who were <30 years old were 94% less likely to have positive attitudes when compared with participants >44 years (aOR = 0.06; *p* < 0.05). It was found that participants working <5 years and between 5 and 10 years were 30 times and 17 times more likely to have positive attitudes towards eye health when compared with participants having >25 years of experience. No other variables were found to be significantly associated with positive attitudes regarding eye health (*p* > 0.05).

### 3.2. Analysis of Practices

Table 5 shows the frequency distribution of practice-related statements. It was found that almost all the participants (95%) prioritise prevention of blindness programmes. More than two-thirds (71%) reported that their spectacle service has a satisfactory turnaround time. Another 70% disagreed that their administration (drug stock/frame stock/IOL stock) is efficiently managed by our ward clerk/s, and 67% indicated that they do not perform noncontact tonometry on all patients. Overall, about three-quarters (73.27%) of the participants were well acquainted with practices on eye health.

Using binary logistic regression, there were statistically significant associations in every category assessed (Table 6). Management were 99% less likely to practice properly towards eye health when compared with RHeH (aOR = 0.012; *p* < 0.01). The Participants Qualified with Certificates and Grade 12 were 92% less likely and participants with postgraduate qualifications were 89% (aOR = 0.106; *p* < 0.01) less likely to know practices related to eye health when compared with participants having a university degree. With regards to age, the middle age group (30–44 years) were about three times more likely to have the best information on practices within the eye clinics (aOR = 2.603; *p* < 0.05) when compared with the >44 years age group. Having <5 years of experience were 27 times more likely to practice properly than those having more than 27 years of experience (aOR = 26.600; *p* < 0.01).

Spearman’s correlation (Table 7) test found significant moderate positive correlation exists between knowledge, attitudes, and practices among the participants.

## 4. Discussion

The study aimed to determine the levels of knowledge, attitudes, and practices of eye health care workers and their supervisors towards eye health. Knowledge, attitude, and practice (KAP) surveys are useful in public health planning, as they collect focused, essential information that is useful in guiding public health programmes [30].

### 4.1. Knowledge on Eye Health

Good knowledge of health is always associated with satisfactory health behaviours and outcome [30]. Therefore, understanding the correlates of good eye health through knowledge leads to improved eye care in a society [31]. The present study found a good level of knowledge among the participants. The study also found that eye health managers had poorer knowledge than the HReH that they supervise. Similarly, other studies conducted in South Africa and Swaziland reported poor knowledge of eye health management, factors attributed to the absence of policies and guidelines on eye health [32,33]. Authors reported a lack of eye health knowledge amongst general practitioners and attributed this to their short training period in this area of healthcare [34]. Other studies that reported reasons for poor knowledge in eye health said that it was due to the fact that it was not a critical “life or death” issue, a lack of adequately trained personnel, a shortage of refresher courses, and that focusing on it would unnecessarily add to their already high workload [35,36]. A recent Ethiopian study found poor knowledge among paediatricians of eye diseases [37].

In this study, education levels were significantly associated with knowledge levels. This finding is similar to that of other studies conducted elsewhere [33,38,39]. These studies showed a correlation between eye health knowledge, age, and the respondents’ education level [31,38]. On the contrary, another study showed no correlation between knowledge of eye health and education level or age [40]. As a result, regardless of how qualified another physician was in another area of health such as orthopaedics, paediatrics, or even general health practice, their knowledge was still poor when it came to eye health. Considering that medical officers and specialists initially qualify as medical doctors, their reported minimal exposure to ocular health in their training is a possible reason for their poor knowledge. As they also spend a few weeks in their ophthalmology block, they do not learn much in this area of health care and as such have poor knowledge in it [39,40,41].

### 4.2. Attitudes towards Eye Health

Health workers who have positive attitudes are more likely to follow standard procedures and apply themselves to their duties, whereas those with negative attitudes would not do the same [42]. In this study, the majority of the participants had positive attitudes regarding eye health. It was also found that the youngest participants had the most negative attitudes. The possible explanation for this is that the youngest participants generally came from the safe and sheltered environment of an academic institution, where there were systems and clear protocols. They had since entered a system that does not have clear processes and guidelines, no dedicated directorate, and no easily available supervision. In addition to the working environment, they generally did not have the basic equipment that they required to perform their basic tasks [24,43]. In realising this, they did not have an understanding supervisor who would realise that urgent procurement of basic equipment was a critical enabler for them to perform their duties. As a result, they found themselves lost. The reality of their internal managers not being trained in eye health, and being incapable of providing clinical guidance and support, might be part of the reason for their negative attitude towards it. In another study, HReH attitudes were far more favourable amongst themselves when they were discussing task sharing as opposed to when they were discussing it with management [44]. A recent study conducted among paediatricians in Jordan reported satisfactory attitudes regarding eye health and disorders [45].

Those who had recently started working in eye health had the most positive attitude compared to those who had been working for more than 15 years. Evidence has shown that even though financial remuneration drives employees, it does not compare to the attainment of certain personal goals, either by progression or vertical promotion [46,47]. Intrinsic drivers include promotion and more responsibility within the employment context, driving better performance, self-actualisation, and job satisfaction in an employee [47,48]. The fulfilment that comes with greater responsibility and decision making often drives millennials (those up to age 40) to work hard as they value climbing the corporate ladder [46,47]. As this is lacking in some areas of HReH employment within DoH, it lowers the employee drive and nurtures a negative attitude towards work. Further to this, the lack of professional support and understanding is a challenge within these eye clinics. Most respondents in this study (78.2%) did not feel that their working space was sufficient for eye health professionals to work in. This is further supported by the majority (89.1%) of respondents who agreed that if directorates want to see positive outputs, they need to provide resources in the eye clinics. Sithole conducted a study among the Directorate Managers and found that there were no guidelines on eye screening, eye protection, and basic eye care [32]. Since the management group generally had a slightly older population, with a long service period, their negative attitude was largely due to a lack of ocular guidelines. They further did not have any ocular directorate at a senior level to look to for guidance, possibly resulting in a negative attitude, and shifted their focus more to their familiar health areas such as geriatrics and NCD [32].

### 4.3. Practices towards Eye Health

Overall, participants were practicing satisfactorily towards eye health. Almost all the participants confirmed that their eye clinics/districts prioritised blindness prevention in their daily practice. This shows that in light of their working circumstances, these eye health workers still aim to practice the highest level of clinical care in their workplaces. Two-thirds (67%) of the participants indicated that eye clinics did not perform tonometry or fundus examinations due to a lack of equipment, showing that the lack of understanding and prioritising of eye health has severe consequences. Another South African study reported that the conventional practice in hospitals is for trainees to perform cataract surgery under the supervision of consultants, and evaluation of the progress in ophthalmic surgical training was essentially an apprenticeship model [47]. To improve cataract surgical outcomes in Africa, “improved training of surgeons” was ranked as the top priority [49,50].

Both managers and HReH were clear about the severe shortage of basic equipment in the eye clinics, as well as the inefficient spectacle supply chain. Furthermore, the replacement of dysfunctional and old equipment is not honoured or prioritised by managers. Another study reported similar findings indicating that South Africa’s primary eye health services lack the organisation and resources to address the leading causes of visual impairment, namely uncorrected refractive error and cataract [23,51,52]. Resource constraints, both human and equipment, are common inhibitors to the delivery of ocular services in African countries [17,52,53]. The shortage of these resources impedes basic practices aimed to ensure the prevention of blindness and PEC. The WHO states that an efficient supply chain and the availability of medicines, and medical devices, are crucial contents in their framework of health systems, to ensure health systems strengthening [8]. This will continue to be unsuccessful if these issues persist as they impede the practices of HReH.

Future studies should seek to include financiers and supply chain managers in public health services, in an effort to understand the details involved in the financing of eye health overall. This will add valuable information and provide further context on the issues raised in this study.

## 5. Conclusions

The overall knowledge, attitudes and practices on eye health were satisfactory among the participants but differed significantly between managers and HReH. It is evident that an appropriate eye health professional should be appointed as part of management at both operational and directorate levels. Resources, both human and equipment, would need to be better allocated by knowledgeable professionals for improvement of clinical practice and eye health services overall. There is a need to review the current management structure, as HReH currently work under difficult conditions.

Despite adding new information to the body of existing knowledge, the limitation of this study was the exclusion of supply chain and finance personnel, who could have given context to the issues that were raised by respondents.

The appointment of a sufficiently trained directorate to manage eye health in each district would be beneficial to eye health and prevention of blindness strategies. This will further ensure that an efficient eye health workforce is placed and managed through optimal governance, resulting in improved eye health outcomes and better service delivery to the communities within the province.

## Figures and Tables

**Figure 1 ijerph-18-12513-f001:**
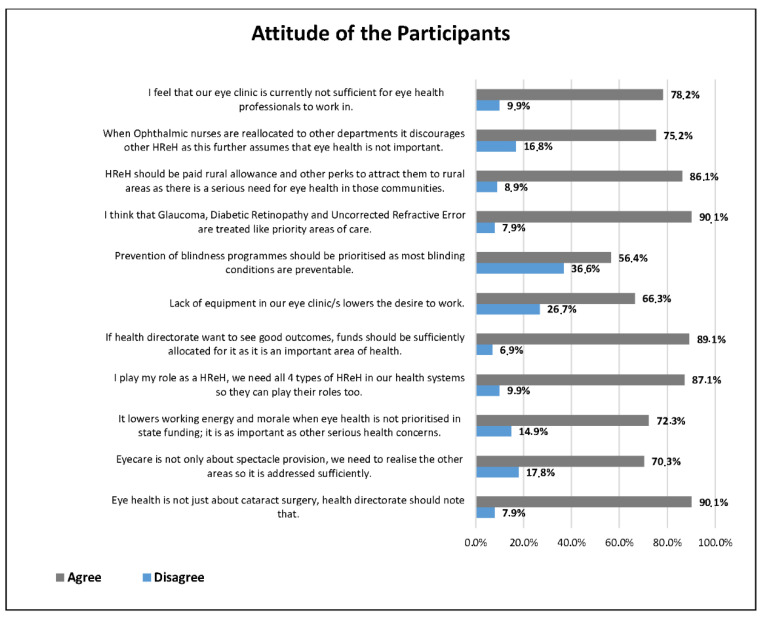
Summary of responses related to attitude towards eye health.

**Table 1 ijerph-18-12513-t001:** Distribution of managers and HReH.

Managers	24	23.76%	HReH	77	76.24%
District Director of NCD (trained as Ophthalmic Nurses)	1	0.99	Ophthalmologists	3	2.97
District Director of NCD (not trained as Ophthalmic Nurses)	4	3.96	OMO	2	1.98
Hospitals CEOs	5	4.95	Optometrists	38	37.62
Medical Managers	13	12.87	Ophthalmic Nurses	24	23.76
Medical Managers/OMO	1	0.99	Nurses	9	8.91
			Eye Clinic clerk	1	0.99

**Table 2 ijerph-18-12513-t002:** Frequency distribution of responses related to knowledge regarding eye health.

Statements	D	N	A
An optician is mainly trained to measure and cut lenses.	13.9	7.9	78.2
An Ophthalmic Nurse provides the role of performing eye screening and assisting in theatre.	11.9	5.0	83.2
An Optometrist is central in performing refraction and low vision services.	11.9	1.0	87.1
An Optometrist is an eye health professional trained through a 4-year university degree.	17.8	5.9	76.2
An Optician is an eye health professional trained through a university of technology diploma.	35.6	17.8	46.5
An Ophthalmologist is an eye health professional trained with a basic medical degree and further training after that.	19.8	73.3	6.9
An Optometrist is the HReH performs general primary eye health.	65.3	3.0	31.7
An Ophthalmologist works in theatre performing eye surgery.	14.9	11.9	73.3
I know which eye health services we provide in my hospital/district/province.	1.0	4.0	95.0
I am fully aware of the programmes that we have in place as a hospital/district/province, in order to assist with prevention of blindness in this region.	5.0	12.9	82.2
In Our district/province we have not yet met the HReH targets in line with the Global Action Plan.	7.9	55.4	36.6
Eye health has not been specified amongst the priority programmes of the NHI.	5.0	41.6	53.5

D = Disagree, N = Neutral, A = Agree.

**Table 3 ijerph-18-12513-t003:** Logistic regression output for having good knowledge.

Variables	Adjusted Odds Radio (aOR)	95%CI		*p*-Value
		Lower	Upper	
Role in Department of Health				
HReH	14.21	1.99	101.28	0.008
Management	1			
Highest Qualification				
Certificate	0.02	0.01	0.38	0.010
Post-basic Diploma	0.09	0.01	1.04	0.054
Postgraduate Qualification	0.07	0.00	1.32	0.076
Undergraduate Diploma	0.35	0.05	2.71	0.315
University Degree	1			
Age				
<30 years	2.36	0.10	58.42	0.599
30–44 years	12.02	2.00	72.09	0.007
>44 years	1			
Period of service				
<5 years	0.21	0.01	5.27	0.344
5–10 years	0.35	0.04	2.85	0.326
11–15 years	0.68	0.08	5.92	0.725
16–20 years	0.80	0.09	7.39	0.846
21–25 years	3.13	0.22	44.47	0.400
>25 years	1			

**Table 4 ijerph-18-12513-t004:** Association between attitude and demographic variables.

Variables	aOR	95%CI		*p*-Value
		Lower	Upper	
Role in Department of Health				
HReH	2.08	0.43	10.03	0.362
Management (ref)	1			
Highest Qualification				
Certificate	0.79	0.08	8.16	0.840
Post-basic Diploma	0.72	0.14	3.63	0.689
Postgraduate Qualification	2.49	0.36	17.44	0.357
Undergraduate Diploma	0.51	0.07	3.65	0.502
University Degree	1			
Age				
<30 years	0.06	0.01	0.98	**0.0411**
30–44 years	0.25	0.04	1.68	0.152
>44 years	1			
Period of service				
<5 years	30.28	1.52	603.24	**0.025**
5–10 years	17.17	2.28	33.94	**0.009**
11–15 years	3.96	0.42	36.87	0.227
16–20 years	4.48	0.48	41.65	0.187
>25 years	1			

**Table 5 ijerph-18-12513-t005:** Practices towards eye health by eye health workers (%).

Statements	D	N	A
Optometrists are restricted to refraction in our hospital/district/province	79 4	7.5	44.6
We perform non—contact tonometry on all patients	67.3	28.7	4.0
We perform a DFE on all chronic patients seen in our clinics	65.3	14.9	19.8
We have equipment that is useable and modern	62.4	22.8	14.9
Our spectacle service has a satisfactory turnaround time	71.3	9.9	18.8
We/our staff have the resources to perform basic slitlamp techniques on all our diabetic patients	62.4	33.7	4.0
We are unable to practice fully in our scopes as we do not have basic equipment for that.	13.9	119	74.3
Our referrals to Ophthalmologists have a turnaround time of up to three weeks	87.1	2.0	10.9
We are aware of prevention of blindness programmes, and we prioritise them in our eye clinic/hospital/district/province.	1.0	4.0	95.0
Our administration (drug stock/frame stock/IOL stock) is efficiently managed by our ward clerk/s	70.3	14.9	14.9
I am satisfied with the district/provincial directorate, as they understand eye health and provide sufficient budgets for it	63.4	15.8	20.8

**Table 6 ijerph-18-12513-t006:** Association between practices and demographic variables.

Variables	aOR	95%CI		*p*-Value
		Lower	Upper	
**Role within Department of Health**				
Management	**0.012**	**0.003**	**0.052**	**<0** **.** **01**
HReH	1			
Highest Qualification				
Certificate and grade 12	**0.083**	**0.013**	**0.544**	**0.009**
Post-basic Diploma	0.708	0.183	2.736	0.617
Postgraduate Qualification	1.000	0.104	9.614	1.000
Undergraduate Diploma	**0.106**	**0.031**	**0.367**	**0.000**
University Degree	1			
**Age**				
30–44 years	**2.603**	**1.006**	**6.737**	**0.034**
>44 years	1			
**Period of service**				
<5 years	**26.600**	**2.626**	**269.409**	**0.005**
5–10 years	**7.560**	**1.700**	**33.629**	**0.008**
11–15 years	2.600	0.598	11.310	0.203
16–20 years	2.100	0.381	11.589	0.395
21–25 years	1.867	0.283	12.310	0.517
>25 years	1			

**Table 7 ijerph-18-12513-t007:** Spearman’s correlation test output.

			Practice	Knowledge	Attitude
Spearman’s rho	Practice	Correlation Coefficient	1.000	**0.499**	0.114
		Sig. (2-tailed)		0.000	0.251
		N	101	101	101
	Knowledge	Correlation Coefficient	**0.499**	1.000	**0.421**
		Sig. (2-tailed)	0.000		0.000
		N	101	101	101
	Attitude	Correlation Coefficient	0.114	**0.421**	1.000
		Sig. (2-tailed)	0.251	0.000	0.000
		N	101	101	101

Correlation is significant at the 0.01 level (2-tailed).

## Data Availability

Data are included in the manuscript.

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
