# Peer review of "Knowledge, Attitudes and Practices of Eye Health among Public Sector Eye Health Workers in South Africa"

_ijerph, 2021, doi:10.3390/ijerph182312513_

Round 1

Reviewer 1 Report

In general, it seems to me to be a very interesting paper with some conclusions that, translated into real life, could significantly improve primary care in eye health and prevention of blindness.

I find the only element to improve the tables, they are unattractive and difficult to read, although they are well explained.

Author Response

Thank you for the kind comments. 

  • Tables look untidy and unclear

They have been amended and somewhat improved.

Reviewer 2 Report

Dear authors, thank you so much for this amazing study. 

I have some recomendations to improve even more your manuscript: 

1) Remove green highlight in line 29. 

2) In the introduction write the objectives of your study in an explicit way. You mention the contributes that your study will give, ok, but it is important to give the list of aims you intend to answer with your study and that will be answered in the results. 

3) In the conclusions write the Weaknesses of your study but also the contributes to clinical practice, to research and to society. 

Author Response

  • Remove green highlight on line 29

Highlight removed.

  • In the introduction, aims and objective should be more explicit.

Aim is expressed in lines 76 – 77.
The word “seeks” was changed to “aims” to make it clearer.
Improved details of aims and objectives of the study

  • In the conclusion write the weakness of your study and contributes to clinical practice, research and society.

This was addressed by adding relevant information to the conclusion.
Minor spell check required Final version of the manuscript has been
read and edited where necessary.

Reviewer 3 Report

The current paper is of interest as it tackles an important topic: the Knowledge, attitudes, and practices of eye health among public sector eye health workers in South Africa. Despite this, some amendments/ modifications need to be made to move forward in the publication process. 

The plagiarism check showed no significant overlap with previously published (18%).

 The authors have performed their study on 110 individuals. How did they get their sample size? Did the authors perform any sampling procedure? How can they be sure that their results are not spurious?

The authors have used LIKERT scaled questionnaires anonymously. However, they did not mention if it was a validated questionnaire .. This point needs to be addressed.

The authors conducted their study in 11 districts of a South African province. Doing some search showed that many official languages exist in South Africa.. did they translate their questionnaire ?? How was translation done?? professional translator ? group members?? 

The authors have written their methods as one part. This structure is unusual. Please consider adding subsections such as; Ethics statements, statistical analysis, questionnaires, inclusion and exclusion criteria.

The introduction section is lengthy and needs shortening.. 

Many sentences in the text, especially in the introduction and the discussion, lack reference(s). The authors need to make sure that every sentence ends with reference(s). 

Author Response

  • Introduction and background.

This has been improved.

  • Methods

This is the only reviewer who queried the methods, so they were left unchanged.

  • Sample size

This is explained in lines 91 – 100 in the manuscript.
From a total population of 194, the study reported on more than 50% which is much higher than the approximate 10 – 15% that most formulae calculate, with a good distribution. This was deemed to be a representative sample as it included all levels of participants from all eleven districts.

  • Plagiarism absent.

Thank you for that confirmation.

  • LIKERT scale and validation

The Likert scale was validated

  • Tool translation due to this being conducted in 11 districts

The districts are in the province of KwaZuluNatal where the dominant language is isiZulu and English. Respondents have been trained in English so their proficiency of the language was good. As they were all trained in this
language, and all work documents are in English, there was no need for translation

  • Adding subsections to methodology.

The authors had originally done this but had to remove subheadings to adhere to journal guidelines.

Reviewer 4 Report

The paper is interesting and offers new information regarding the knowledge and practices of eye health among public sector eye health workers in South Africa. There are some issues that need to be addressed:

  1. Figure 1 is duplicated and the text of some questions is incomplete (information is missing)
  2. the distribution of the interviewed healthcare personnel according to their qualification would be interesting for the readers

Author Response

  • Figure 1 is duplicated and the text of some questions is incomplete (information is missing)

They have been amended and somewhat improved.

  • the distribution of the interviewed healthcare personnel according to their qualification would be interesting for the readers

Thank you for the constructive input. 

Round 2

Reviewer 3 Report

The authors answered my comments